# Presenting Symptoms in Newly Diagnosed Myeloma, Relation to Organ Damage, and Implications for Symptom-Directed Screening: A Secondary Analysis from the Tackling Early Morbidity and Mortality in Myeloma (TEAMM) Trial

**DOI:** 10.3390/cancers15133337

**Published:** 2023-06-25

**Authors:** Stella Bowcock, Catherine Atkin, Gulnaz Iqbal, Guy Pratt, Kwee Yong, Richard D. Neal, Tim Planche, Kamaraj Karunanithi, Stephen Jenkins, Simon Stern, Sarah Arnott, Peter Toth, Farooq Wandroo, Janet Dunn, Mark T. Drayson

**Affiliations:** 1Department of Haematological Medicine, King’s College Hospital NHS Trust, London SE5 9RS, UK; 2Princess Royal Hospital, King’s College Hospital NHS Trust, Orpington Common, London BR6 8ND, UK; 3Institute of Inflammation and Ageing, University of Birmingham, Edgbaston, Birmingham B15 2GW, UK; 4Warwick Clinical Trials Unit, University of Warwick, Coventry CV4 7AL, UK; 5Queen Elizabeth Hospital, University Hospitals Birmingham NHS Trust, Birmingham B15 2TH, UK; 6Department of Haematology, UCL Cancer Institute, London NW1 2BU, UK; 7Department of Primary Care Medicine, University of Exeter, Exeter EX1 2LU, UK; 8Department of Medical Microbiology, St George’s Hospital NHS Trust, London SW17 0QT, UK; 9Royal Stoke University Hospital, University Hospitals North Midlands NHS Trust, Stoke-on-Trent ST4 6QG, UK; 10Russell Halls Hospital, The Dudley Group NHS Foundation Trust, Dudley DY1 2HQ, UK; 11Epsom and St Helier NHS Trust, London SM5 1AA, UK; 12Medway NHS Trust, Gillingham ME7 5NY, UK; 13Sheffield Teaching Hospitals NHS Trust, Sheffield S10 2JF, UK; 14Sandwell General Hospital, Lyndon, West Bromwich, West Midlands B71 4HJ, UK; 15Institute of Immunology and Immunotherapy, University of Birmingham, Birmingham B15 2TT, UK

**Keywords:** multiple myeloma, symptoms, organ damage, screening, precursor disorder, MGUS, smouldering myeloma, diagnosis, diagnostic delay

## Abstract

**Simple Summary:**

Myeloma, a blood cancer, is rare and hard to diagnose. People often suffer irreversible organ damage by the time of diagnosis. Myeloma is preceded by a premalignant phase that is easily identifiable on a blood test. Currently, there is no screening for this, because most people do not progress to myeloma. We aimed to inform guidelines and screening by refining our understanding of how patients developing myeloma describe their symptoms and how those symptoms relate to organ damage. We found that patients rarely describe ‘bone pain’ but simply ‘pain’. Low-impact crush fractures of the backbones appear to be under-recognised as abnormal. At least 30% of patients have irreversible organ damage at diagnosis. People who developed myeloma fared better if they had previously been diagnosed to have the premalignant condition. Screening based on certain symptoms, possibly combined with imaging and laboratory results, may speed up the diagnosis of myeloma.

**Abstract:**

Multiple myeloma (MM) patients risk diagnostic delays and irreversible organ damage. In those with newly diagnosed myeloma, we explored the presenting symptoms to identify early signals of MM and their relationships to organ damage. The symptoms were recorded in patients’ own words at diagnosis and included diagnostic time intervals. Those seen by a haematologist >6 months prior to MM diagnosis were classified as precursor disease (PD). Most (962/977) patients provided data. Back pain (38%), other pain (31%) and systemic symptoms (28%) predominated. Patients rarely complain of ‘bone pain’, simply ‘pain’. Vertebral fractures are under-recognised as pathological and are the predominant irreversible organ damage (27% of patients), impacting the performance status (PS) and associated with back pain (odds ratio (OR) 6.14 [CI 4.47–8.44]), bone disease (OR 3.71 [CI 1.88–7.32]) and age >65 years (OR 1.58 [CI 1.15–2.17]). Renal failure is less frequent and associated with gastrointestinal symptoms (OR 2.23 [CI1.28–3.91]), age >65 years (OR 2.14 [CI1.28–3.91]) and absence of back pain (OR 0.44 [CI 0.29–0.67]). Patients with known PD (*n* = 149) had fewer vertebral fractures (*p* = 0.001), fewer adverse features (*p* = 0.001), less decline in PS (*p* = 0.001) and a lower stage (*p* = 0.04) than 813 with de novo MM. Our data suggest subgroups suitable for trials of ‘symptom-directed’ screening: those with back pain, unexplained pain, a general decline in health or low-impact vertebral compression fractures.

## 1. Introduction

Multiple myeloma (MM) is a cancer of plasma cells that causes bone lesions, hypercalcaemia, anaemia, renal failure and susceptibility to infection. The rate of survival has improved dramatically, but myeloma patients experience some of the longest times to diagnosis of all cancers, with a median diagnostic interval in the UK of 163 days and similar delays in other countries [1,2,3,4,5]. Patients may experience irreversible organ damage by the time of presentation with consequent significant morbidity, possibly from diagnostic delay [6,7].

The symptoms of myeloma are usually non-specific and overlap with other common conditions, giving a low positive predictive value (PPV) for any one presenting symptom [2,8,9]. In primary care, it is deemed ‘harder to suspect’ than most other cancers, compounded by its rarity, comprising only 2% of all cancers [10,11]. Paradoxically, myeloma and its precursor diseases have an accessible screening test in the form of a ‘myeloma screen’ blood +/− urine test. However, the challenge, in the absence of a screening programme, is for healthcare workers to even consider the diagnosis. Population studies have suggested that those with a previously identified precursor plasma cell disease (PD) of monoclonal gammopathy of undetermined significance (MGUS) or smouldering myeloma (SMM) have fewer major complications at active myeloma diagnosis and better overall survival than those presenting de novo [12,13]. The current ‘Iceland screens, treats or prevents multiple myeloma’ (iStopMM) trial aims to assess the benefits and risks of whole population screening to diagnose plasma cell disorders [14]. We aimed to review the presenting symptoms of newly diagnosed myeloma patients and their development during the diagnostic path, including those progressing from a known PD, to inform guidelines, screening and improve the speed of diagnosis. We also explored the relationship between symptoms and irreversible organ damage and compared these between PD patients and those with de novo MM. Data on the diagnostic pathways for this patient cohort are published elsewhere [15].

## 2. Materials and Methods

### 2.1. Data Collection

The TEAMM trial (Tackling EArly Morbidity and Mortality in Myeloma) was a randomised, double-blinded, placebo-controlled trial of levofloxacin in the first 12 weeks after an active MM diagnosis (ISRCTN51731976) to assess whether levofloxacin reduced febrile episodes and deaths and its effects on healthcare-related infections [16]. The secondary endpoints included factors that may influence the prognosis and susceptibility to infections. Ethical approval was provided by NHS Research Ethics Committee West Midlands and sponsorship by the Universities of Birmingham and Warwick. The inclusion criteria were broad, with patients required to have newly diagnosed active multiple myeloma according to the International Myeloma Working Group (IMWG) criteria (later updated), able to give informed consent and within 14 days of starting anti-myeloma treatment (Appendix A) [17,18].

The date of trial entry was taken as the date of active MM diagnosis. Data were collected from individual participants at trial entry, with answers based on patient recall during a single interview. Patients were asked to list the symptoms they attributed to their myeloma, along with the date of onset of each symptom. The symptoms were recorded text-free from the patient’s own words. The number of times and dates they visited their primary care physician before diagnosis, the date and hospital department they first visited due to their myeloma symptoms and when they first saw a haematologist were also collected. The diagnostic interval questions were modified after 197 patients to align with the Aarhus criteria (Appendix A) [19]. Patients were asked their current Eastern Cooperative Oncology Group (ECOG) performance status (PS) and PS at 6 months prior to diagnosis.

Patients who started treatment 6 months or more after their first haematology appointment or explicitly stated among their patient-reported symptoms that they initially had PD were classified as having probable PD. Other patients were designated ‘de novo’ myeloma.

The organ damage parameters explored were CRAB features (CRAB criteria = hypercalcaemia, renal failure, anaemia and bone lesions) and vertebral fractures [17]. Hypercalcemia, renal failure and anaemia as the CRAB features were determined from blood test results available at diagnosis based on IMWG criteria. Vertebral fractures were used as a measure of established irreversible organ damage at diagnosis, as was spinal cord compression. The presence of vertebral fracture, bone lesions and spinal cord compression included those reported by patients or recorded from imaging reports at trial entry. Renal failure (estimated glomerular filtration rate <40 mL/min) was assessed separately as renal damage could be due to multiple pathologies and may also reverse with therapy. The number of patients with potential myeloma-induced irreversible organ damage was taken to be those where their estimated glomerular filtration rate remained <40 mL/min at 12 months after trial entry, excluding those on diabetic medication, where diabetic renal disease could have been contributory.

### 2.2. Analysis of Symptoms

Symptoms were grouped independently into categories by 3 clinicians according to how a healthcare professional might perceive the symptoms or incidental blood results and aligned with previous studies [2,8,9]. An incidental abnormality on a routine or screening blood test was included as a ‘symptom’ if it triggered the pathway to diagnosis but documentation of myeloma confirmatory tests were not included. The first detectable symptoms and final total symptoms at trial entry were grouped. The first detectable symptoms were those described at, or prior to, the first presentation to any healthcare professional, reflecting the symptoms that were available to prompt the initial diagnostic investigations.

### 2.3. Statistical Analysis

The number of symptoms, association of symptoms with the baseline parameters, organ damage, decline in PS and ISS and comparison of patients with a prior diagnosis of precursor disease versus ‘de novo’ myeloma were explored using the chi-square test for categorical data (the continuity adjustment, Fisher’s exact and Mantel–Haenszel tests used as appropriate) and Wilcoxon rank-sum tests for continuous data. Missing data were excluded from all tests.

Logistic regression models were used to ‘predict’ patients with vertebral fractures and renal failure. All available symptom data, as well as age, sex, and ethnicity (White versus non-White), were considered in both analyses. Forward, backward and stepwise logistic regression analyses were carried out using a 5% significance level to determine the independent factors for the prediction of those with irreversible organ damage. Regression coefficients, *p*-values and odds ratios are presented. The model accuracy was calculated, and the sensitivity, specificity, proportion of false positives and negatives and overall percentage of correct predictions are presented. The probability cut point was chosen to balance the sensitivity and specificity. All data were analysed using SAS statistical software version 9.4 (SAS Institute, SAS Circle, Cary, NC, USA).

### 2.4. Patient Involvement

Patient representatives and Myeloma UK were involved in the initial TEAMM trial design. Patients were sent the manuscript in the early and late draft stages and their comments incorporated.

## 3. Results

The trial was conducted in 93 UK hospitals between August 2012 and April 2016, with 977 patients entered. Data on the symptoms were available for 962 patients (Appendix A). The baseline characteristics are shown in Table 1, grouped by whether patients reported bodily symptoms. The median age was 67 years (IQR: 60–75), and 74% were PS 0–1. Those who did not report bodily symptoms (67/962, 7%) had a better performance status at diagnosis and were less likely to have skeletal disease compared to those with symptoms but had similar disease stages and rates of anaemia and renal impairment. 

### 3.1. Symptom Profile

Categorisation of the 114 different individual symptoms/reasons for referral is shown in Appendix A. Table 2 shows the number of patients with symptoms within each category, both at the first presentation and in total by the time of trial entry. Many (766) patients reported a total of 1216 symptoms prior to the first consultation with a healthcare worker (HCW). By the time of diagnosis, 881 patients reported 1656 symptoms, with a median of 1 symptom (range 0–6). Thirty-two patients (3%) specified that their diagnosis was an incidental finding.

The commonest symptom categories were back pain (38% of patients), other pain (31%) and general systemic symptoms (28%) (Table 2). These three categories were markedly more frequent than the next commonest category, self-reported anaemia (12%), as a reason for referral. In order of frequency, ‘other pain’ comprised chest/rib, generalised, hip, shoulder and leg pain (Table 2 and Appendix A). The commonest systemic symptoms were fatigue (16%) and weight loss (9%). Only 4% of all patients used the phrase ‘bone pain’ but instead described pain in a particular part of the body or generalised. Fracture or incidental bone pathology detected on imaging was reported as a symptom in only 4% of patients, including twenty patients (2%) reporting a fracture as a symptom. Imaging-recorded fractures/vertebral compressions were present at trial entry for 356 patients (35%) (Appendix A). Spinal cord compression was reported by 2 patients, with 23 reporting possible compression symptoms (0.2–2%). Sixty-eight (7%) stated anaemia as a symptom at the first presentation. At diagnosis, 113 patients (12%) reported anaemia as a contributory symptom at any point in the pathway. Of these patients, 86 had a haemoglobin result available, 56/86 (65%) met the IMWG criteria for anaemia and 30 (35%) did not [17].

The number of patients and total number of symptoms reported increased from the first symptom presentation to trial entry/active MM diagnosis (Table 2). This suggests patients continued to deteriorate during the diagnostic pathway. Accounting for comorbidities did not affect the overall results, since only 1.9% of comorbidities were severe or very severe and unrelated to myeloma.

### 3.2. Relationship between Symptoms and Baseline Parameters

#### Symptom Burden

Compared to those with no symptoms, those with symptoms were more likely to have a decline in PS (*p* = 0.001), vertebral fractures (*p* = 0.008), bone disease (*p* = 0.001) and ≥1 CRAB features (*p* = 0.01) but not anaemia or renal failure (Table 1). These data suggest that bone disease and fractures are the greatest drivers of symptoms.

### 3.3. Symptoms and Their Relationship to Organ Damage, Decline in PS and ISS

Assessing the three commonest symptom categories of ‘back pain’, ‘other pain’ and ‘systemic symptoms’ using a univariate analysis, those with ‘back pain’ were more likely to have vertebral fractures and bone disease. ‘Other pain’ was associated with the presence of bone disease, but those patients were less likely to have vertebral fractures. ‘Systemic symptoms’ were associated with anaemia (Bonferroni-corrected *p* = 0.004 for all factors) (Appendix A).

Table 3 shows that a decline in PS was associated with ‘back pain’ but not ‘other pain’ or ‘systemic symptoms’ and was also associated with vertebral fractures, bone disease, anaemia and poorer ISS (*p* = 0.02).

Exploring the association between the symptoms and potential irreversible organ damage (Appendix A) showed that those with vertebral fractures were more likely to have back pain (*p* = 0.002) and less likely to have systemic symptoms (*p* = 0.04), self-reported anaemia (*p* = 0.009) and respiratory symptoms (*p* = 0.02). There were too few patients with confirmed spinal cord compression (2) for a meaningful analysis, but all had vertebral fractures and were thus included in the analysis. Those with renal failure (eGFR <40mL/min) at trial entry were more likely to have gastrointestinal symptoms (*p* = 0.01). Of the 148 patients with renal failure at trial entry, 33 (3% of total patients) had renal failure unchanged at 12 months, excluding 9 patients on diabetic medication. This underestimated the irreversible renal failure rates, as data on eGFR at 12 months were not available in 52/148 (35%) with a baseline eGFR <40 mL/min versus 173/800 (21.6%) with eGFR >40 mL/min (i.e., data were not missing at random). At minimum, the total irreversible organ damage at an active MM diagnosis was 30% (vertebral fractures in 27%, and proven irreversible renal failure in 3%).

Logistic regression (943 patients, including 270 with vertebral fractures) identified back pain (odds ratio (OR) 6.14 [CI 4.47–8.44]), bone disease (OR 3.71 [CI 1.88–7.32]) and age >65 years (OR 1.58 [CI 1.15–2.17]) as the most important predictors of a vertebral fracture (Table 4). Logistic regression (948 patients, including 148 with renal failure) identified gastrointestinal symptoms (OR 2.23 [CI 1.28–3.91]) and age >65 years (OR 2.14 [CI 1.28–3.91]) as positive predictors and back pain (OR 0.44 [CI 0.29–0.67]) or other pain (OR 0.55 [CI 0.36–0.85]) as significant negative predictors of renal failure.

### 3.4. Comparison of Patients with a Prior Diagnosis of Precursor Disease Versus ‘De Novo’ Myeloma

One hundred and forty-nine individuals with PD were identified, with a median of 1.64 years (IQR 0.91–3.34) reported by these patients between the first haematology appointment and diagnosis of active MM. PD patients showed fewer symptoms than the de novo group (Table 1 and Table 5), although the profile of initial symptoms was similar in both groups. The PD patients had lower rates of bone disease (*p* = 0.001) and anaemia (*p* = 0.03) and fewer CRAB features (*p* = 0.001) and vertebral fractures (*p* = 0.001) (Table 6). Although there was no difference in PS 6 months before trial entry, fewer PD patients reported a decline in PS (*p* = 0.001), although there was no difference in PS between the groups at 6 months before trial entry. The overall survival (OS) was high at 12 months at 91%, and therefore, subgroup analysis was not widely explored. The OS at 12 months in the PD versus the de novo groups were 94% [95% CI = 87–97] versus 90% [95% CI = 87–92], respectively (*p* = 0.15).

## 4. Discussion

This is the largest study to date of presenting symptoms in myeloma recorded from patients’ own words at diagnosis. Our results show that patients rarely complain of ‘bone pain’ but simply ‘pain’ in the back, elsewhere or generalised. Vertebral fractures are under-recognised as pathological, and ‘fatigue’ may be underappreciated. A decline in PS and its relationship to the symptoms and organ damage have never previously been explored in a large study. The symptom burden is high and dominated by pain from bone disease. Vertebral fractures are the predominant irreversible organ damage, directly impact PS, and are associated with back pain and increasing age. Renal failure is less frequent and associated with gastrointestinal symptoms, increasing age and the absence of back pain. Previously diagnosed PD patients who progress to active MM have fewer symptoms, better baseline parameters, including ISS, and less organ damage and PS decline than those diagnosed de novo with MM.

### 4.1. Profile of Patient Symptoms: Patients Rarely Complain of ‘Bone Pain’

Our data clarify how patients describe their symptoms, whereas the previous literature has often used healthcare worker-chosen categories (Table 7). The symptoms where there is the highest divergence between published studies are bone pain, pain and fatigue. Our observation that 89% of patients in the ‘other pain’ category did not complain of ‘bone pain’ concurs with other studies that recorded symptoms in the patients’ own words or as categorised during primary care before the diagnosis was known [2,8,9]. These studies found many patients with ‘pain’ but few describing ‘bone pain’. This contrasts with studies that used retrospective healthcare record reviews [7,20] or predetermined categories chosen by specialists [21] that recorded more patients with ‘bone pain’. The phrase ‘bone pain’ may represent the projection of healthcare worker views on patient symptoms. Qualitative interviews suggest that retrospective attribution may occur [22]. Recognising that patients may complain simply of ‘pain’, either localised or generalised, is key to focusing guidelines and the early recognition of active MM.

Only four (0.4%) of all patients reported vertebral fractures as a trigger for referral, although 27% were recorded in the baseline data to have an imaging report with a vertebral compression fracture or collapse. This suggests that the concept of a ‘vertebral/back fracture’ may be under-recognised as ‘pathological’, either by the patient or referring physician, as previously noted by the Osteoporosis Society [25]. Fatigue/tiredness was the commonest systemic symptom (16%) and described previously in 43% and 32% of patients at diagnosis [2,20]. It affects 99% of all patients at some time during the course of myeloma [26] but is not recorded as a significant symptom in the primary care Clinical Practice Research Datalink (CPRD) [9], suggesting that it may be under-recognised [27].

Our symptom profile (Table 2 and Appendix A) is non-specific and, apart from bone pathology (4%) and incidental blood results suggestive of myeloma (5%), overlaps with other, commoner medical conditions. This problem is accentuated, as myeloma is more common in older individuals when comorbidities may obscure emerging myeloma symptoms [28]. Back pain, when first presenting to a primary physician, has a positive predictive value (PPV) for myeloma of 0.1%, rising to 0.2% at the second visit [9], and a <1% risk that it has a malignant underlying cause [29]. Weight loss (9% of patients) might be regarded as an alarm symptom, yet it only has a PPV for myeloma of 0.2% [9]. Thus, a long diagnostic pathway with progressive organ damage will inevitably continue for some patients unless some form of screening is implemented. In our cohort, only 3% of patients (32/962) had no symptoms prior to the first presentation, with the investigation initially triggered by an abnormal test result.

Where anaemia was reported as a symptom prior to referral, 35% did not actually have anaemia sufficient to meet the IMWG criteria [17,18]. One of the early indicators of progression to myeloma in primary care is a fall in haemoglobin below the previous baseline (but less than the IMWG diagnostic criteria) and an increase in the inflammatory marker ESR or plasma viscosity [30].

### 4.2. Symptoms and Relationship to Organ Damage

A relationship between the symptoms and organ damage is inevitable, because the definition of active symptomatic MM, until 2014, required the demonstration of end organ damage [17]. Only 0.7% of our patients were defined as active MM by the biomarker criteria [18]. Yet, there are varying degrees of end organ damage, some irreversible, with fractures and renal failure affecting the prognosis [31,32]. We show a high level of irreversible organ damage at diagnosis (at least 30%), and considering our patient median age was 67, compared with 70 for the population data, this may be higher in a real-world population [28].

The symptom profiles at the first symptom detection and at the time of diagnosis are almost identical, although numerically less frequent. The frequency of back pain as the initial symptom and the strong relationship between this and vertebral fractures suggests that vertebral fractures are the first presenting symptom in some patients. Nine percent of asymptomatic patients already had vertebral fractures (Table 1), although some may have possibly experienced a previous back pain episode not attributed to myeloma. Even with close PD follow-up, vertebral fractures cannot be completely prevented [33]. Patients broadly present with either symptoms of bone disease with pain or systemic symptoms.

### 4.3. Comparison between Patients with De Novo Myeloma versus Those with a Previous Diagnosis of PD

PD patients fared better by all measures than those presenting with de novo MM. The MM patients almost invariably passed through a precursor phase [34]. Inevitably, our PD cohort showed a selection bias, since they presented either with symptoms or a comorbidity that led to a PD diagnosis. Either the PD cohort was similar biologically to the de novo MM cohort but was diagnosed earlier in their progression to MM, or the PD cohort had an innately less aggressive disease. The latter cannot be excluded, although the monitoring time course of the PD cohort (median 1.64 years) compared to that of a screening population transitioning to active MM (median 3.35 years) favours our PD cohort having a more aggressive disease [35]. Alternatively, our PD cohort may have been further along the path of evolution to MM compared to a population screening cohort. This theory was supported in that our PD cohort showed a similar initial presenting symptom profile, though numerically less frequent, to that of the ‘de novo’ MM cohort, suggesting that symptoms in some were beginning to emerge (Table 5). Despite some selection bias in our PD cohort, the comparison was helpful and reflected the real-world experience of monitoring PD in 93 hospitals in the UK. Notably, our PD cohort developed a significant increase in symptomatology before active MM diagnosis, suggesting that the current PD follow-up might not be optimal.

### 4.4. Implications for Practice

#### 4.4.1. Symptom Profile and Changes to Guidelines

Our results suggest the need to refocus national and international guidelines and the education of healthcare workers concerning the profiles of presenting symptoms of active MM and in monitoring PDs.

#### 4.4.2. Benefits of Detecting Those with PD

Our findings support the need to diagnose plasma cell disorders in the PD phase. The publication of two phase 3 trials showing a benefit in terms of progression-free survival [36,37] and, in one study, overall survival [36] for treatment in high-risk SMM supports the need to identify patients with SMM. The Icelandic iStopMM study (NCT 03327597) might show a benefit to population screening, but this might not be feasible in all countries [12,14,38]. Targeted screening also needs to be explored. One approach is to screen patients who present to healthcare workers with early symptoms of back pain, unexplained pain or deterioration in health. Although these symptoms are common in primary care and, hence, screening may demonstrate a low incidence of active MM, this strategy may prove more specific than whole population screening and identify a population who are in the early stages of progression from SMM to MM. For example, the iStopMM study demonstrated four cases of undiagnosed active MM in a screened population of 75,422 individuals [39], whereas data from the CPRD suggested the incidence of MM in those at their first consultation for back pain was 1 per 1000 [9]. It would also likely identify approximately 40 patients with MGUS [40]. Targeting those developing early symptoms may not eliminate irreversible organ damage but may reduce it and prove cost-effective and reduce overdiagnoses. In addition, radiology reporting could recommend myeloma screening in vertebral compression fractures or osteopenia/osteoporosis. Previous reports suggest that screening those referred to a fragility fracture liaison or osteoporosis service has revealed undiagnosed MGUS or MM [41,42,43,44,45]. Other targeted screening includes those of certain ethnic groups, relatives of those with MM and acute medical hospital admissions [46,47,48]. Indirect approaches might include mandating laboratory automatic myeloma screening in patients with high globulins or the profound suppression of immunoglobulins and those with unexplained anaemia. Focused laboratory comments to aid clinician interpretations of results may also be helpful. Algorithms might trigger myeloma screening in those with certain symptoms combined with laboratory results [49]. Importantly, any screening approach will require a prospective study that includes a heath economic assessment. Wider testing could impact haematologists with increasing referrals, though an algorithm that differentiates patients with MGUS from myeloma with 98% sensitivity may help the triage [50].

#### 4.4.3. Morbidity of Bone Disease

The greatest morbidity at diagnosis is due to bone disease, and the association of vertebral fractures with a decline in PS indicates a subgroup which may benefit from enhanced supportive care, such as the appropriate use of bisphosphonates, good analgesia and a physical activity programme [51].

### 4.5. Strengths and Limitations

The strengths of this study are the large number of patients, contemporaneous data collection, capturing the patients’ own words and information on those who were asymptomatic with an incidental diagnosis. Another strength is a clean dataset of patients with active MM requiring treatment compared to registry studies that do not distinguish between smouldering and active myeloma (Table 7) [2,8,9].

A limitation of our study is that the trial population recruited was younger (median 67 versus 70 years) than expected from the population data [28]. Patients who enter trials may be more willing to seek medical care and represent an earlier symptomatic group than a real-world myeloma population. Symptom data and dates were based on patient recall and may have suffered from omission or inaccuracy, especially the underestimation of time intervals. The total irreversible organ damage was underestimated, since it did not include those with non-spinal fractures who may have suffered from an incomplete return of function. Bone disease and vertebral fracture were strongly associated with back pain and might possibly trump other symptoms, causing the underreporting of less prominent symptoms. Nevertheless, the symptoms reported represented what patients thought was of the most importance.

## 5. Conclusions

At a multiple myeloma diagnosis, the symptom burden is high, dominated by pain, and irreversible organ damage is frequent. Unfortunately, myeloma symptoms are non-specific, and therefore, a long diagnostic pathway is likely to continue if we rely upon healthcare workers to think of the diagnosis before a myeloma screen is performed. Our data support the need to identify those with a precursor disease by some form of screening. Trials of screening using algorithms combining symptoms and/or imaging or laboratory changes may improve the speed of a myeloma diagnosis.

## Figures and Tables

**Table 1 cancers-15-03337-t001:** Number of reported bodily symptoms in relation to the prognostic factors and CRAB features.

		No Reported Bodily Symptoms N = 67		Symptoms Reported N = 895		*p* *
Factor	Grouping	N	%	N	%	
Age (years)	Median	67		67		0.84
	IQR	64–77		59–75		
Ethnicity (White versus non-White)	White	64	95	812	91	>0.99
Gender	Male	44	66	557	62	>0.99
Planned high intensity treatment	Yes	31	46	492	55	>0.99
PS 6 months before diagnosis MM	0–1	63	94	815	91	>0.99
	2–4	2	3	48	5	
	Unavailable	2	3	32	4	
PS at diagnosis of MM	0–1	61	91	663	74	0.01
	2–4	6	9	214	24	
	Unavailable	0	0	18	2	
Decline in PS	No change	56	84	480	54	0.001
	Deteriorated	9	13	382	43	
	Unavailable	2	3	33	4	
ISS	1	17	25	196	22	>0.99
	2	28	42	322	36	
	3	13	19	234	26	
	Unavailable	9	14	143	16	
Hypercalcaemia	Yes	3	5	42	5	>0.99
	No	53	79	655	73	
	Unavailable	11	16	198	22	
Renal impairment	Yes	12	18	136	15	>0.99
	No	54	81	746	83	
	Unavailable	1	1	13	1	
Anaemia	Yes	23	34	376	42	>0.99
	No	35	52	344	38	
	Unavailable	9	14	175	20	
Bone disease	Yes	29	43	651	73	0.001
	No	37	55	238	27	
	Unavailable	1	2	6	<1	
Vertebral fracture	Yes	6	9	264	30	0.008
	No	21	31	377	42	
Number of CRAB features	0	11	16	63	7	0.01
	1	30	45	338	38	
	2	10	15	232	26	
	3+	3	5	59	7	
	Unavailable	13	19	203	23	
Precursor disease	Yes	31	46	118	13	0.02
	No	36	54	777	87	

* *p*-values with Bonferroni correction [14]. PS = Eastern Cooperative Oncology Group (ECOG) performance status (PS); MM = multiple myeloma; ISS = International Staging System, International Myeloma Working Group (IMWG) definition of hypercalcaemia = serum calcium > 2.75 mmol/L; renal impairment = creatinine clearance <40 mL/min (post-hydration baseline estimated the glomerular filtration rate used in this study); bone = any myeloma-defining bone lesion on imaging; vertebral fracture = baseline imaging included a report of vertebral fracture/collapse and CRAB features = hypercalcaemia, renal failure, anaemia and bone lesions.

**Table 2 cancers-15-03337-t002:** Number of patients reporting each symptom. They were grouped as the first presenting symptoms or abnormalities triggering investigation and those present at diagnosis. Percentages are shown as the proportion of TEAMM participants with data available for questions on symptoms prior to diagnosis (*n* = 962). The commonest subgroups within each category from this table are included in italics.

Categories of Self-Reported Symptoms and Other Reasons for Referral	Initial Symptoms or Abnormality Reported N = 962		Total Symptoms or Abnormalities Reported at MM Diagnosis N = 962	
	N	%	N	%
Back pain	301	31	364	38
Other pain (*excluding back pain*)	234	24	295	31
*Chest pain*	*80*	8	*97*	10
*Pain (general)*	*59*	6	*73*	8
*Hip pain*	*40*	4	*53*	6
*Shoulder pain*	*26*	3	*39*	4
Systemic symptoms	208	22	265	28
*Fatigue*	*122*	13	*155*	16
*Weight loss*	*72*	7	*89*	9
Self-reported anaemia	69	7	113	12
Respiratory symptoms	59	6	74	8
Gastrointestinal symptoms	53	6	73	8
*Abdominal pain*	*26*	3	*35*	4
*Nausea & vomiting*	*17*	2	*23*	2
Infection	42	4	54	6
*Lower respiratory tract infection* *	*23*	2	*31*	3
Neurological symptoms	33	3	53	6
Abnormal blood results (suggestive of myeloma)	29	3	51	5
Self-reported renal problems	21	2	46	5
Other abnormal blood results	21	2	32	3
Other symptoms	19	2	23	2
Bleeding/thrombosis	16	2	24	2
Lump	9	1	12	1
Urological symptoms	6	1	8	1

* Includes single and recurrent infections.

**Table 3 cancers-15-03337-t003:** Decline in the ECOG performance status (≥1 point) prior to diagnosis and relation to end organ damage and prognostic features at trial entry for 927 patients with data available at both timepoints (* *p*-values with Bonferroni correction).

		Change in ECOG Group						
Factor	Grouping	Improved/No Change (*n* = 25/511)		Deteriorated *n* = 391		Total *n* = 927		*p* *
		N	%	N	%	N	%	
Gender	Female	199	37	141	36	340	37	>0.99
	Male	337	63	250	64	587	63	
Ethnicity	White	483	90	359	92	842	91	>0.99
	Other	53	10	31	8	84	9	
Age (years)	N	536		391		962		>0.99
	Median	67		67		67		
	IQR	60–75		60–75		60–75		
Back pain at diagnosis	No	360	67	210	54	570	61	0.002
	Yes	176	33	181	46	357	39	
Other pain at diagnosis	No	384	72	261	67	645	70	>0.99
	Yes	152	28	130	33	282	30	
Systemic symptoms at	No	399	74	269	69	668	72	>0.99
diagnosis	Yes	137	26	122	31	259	28	
Any fracture	No	379	71	201	52	580	63	0.002
	Yes	153	29	187	48	340	37	
Vertebral fracture	Yes	109	20	146	37	255	28	0.002
eGFR group	<40 mL/min	79	15	67	17	146	16	>0.99
	40+ mL/min	454	85	318	83	772	84	
ISS	I	130	30	75	22	205	26	0.36
	II	189	43	151	44	340	43	
	III	121	28	119	34	240	31	
Hypercalcaemia	Absent	404	96	279	92	683	94	0.72
	Present	18	4	25	8	43	6	
Renal impairment	Absent	394	91	266	86	660	89	>0.99
	Present	40	9	42	14	82	11	
Anaemia	Absent	242	55	119	39	361	49	0.002
	Present	195	45	188	61	383	51	
Bone disease	Absent	193	36	74	19	267	29	0.002
	Present	339	64	314	81	653	71	
Number of CRAB	0	59	14	14	5	73	10	0.002
features	1	229	55	126	42	355	49	
	2	111	27	121	40	232	32	
	3+	18	4	41	14	59	8	
Precursor disease	No	430	80	356	91	785	85	0.002
	Yes	106	20	35	9	141	15	
Number of symptoms	0	127	24	43	11	170	19	0.002
at diagnosis	1	239	46	202	52	441	48	
	2	100	19	97	25	197	22	
	3	42	8	33	9	75	8	
	4+	16	3	12	3	28	3	

**Table 4 cancers-15-03337-t004:** Multivariate tables for the risk of developing irreversible organ damage at diagnosis.

Risk Factor	Parameter Estimate	Standard Error	*p*-Value	Odds Ratio	95% CI
1.a Vertebral fractures, N = 943, event = 270					
Intercept	−2.54	-	-	-	-
Back pain (absent versus present)	1.82	0.16	<0.0001	6.14	4.47–8.44
Bone disease/non-vertebral fracture (absent versus present)	1.31	0.35	0.0002	3.71	1.88–7.32
Age group (≤65, >65 years)	0.46	0.16	0.005	1.58	1.15–2.17
1.b Vertebral fractures with fall in ECOG considered, N = 908, events = 255					
Intercept	−2.97	-	-	-	-
Back pain (absent versus present)	1.82	0.17	<0.0001	6.17	4.43–8.61
Fall in ECOG (absent versus present)	0.79	0.17	<0.0001	2.19	1.58–3.05
Bone disease/non-vertebral fracture (absent versus present)	1.17	0.36	0.001	3.23	1.60–6.54
Age group (≤65, >65 years)	0.43	0.17	0.01	1.54	1.11–2.15
2.a eGFR < 40, N = 948, events = 148					
Intercept	−2.55	-	-	-	-
Back pain (absent versus present)	−0.82	0.21	0.0001	0.44	0.29–0.67
Age group (≤65, >65 years)	0.76	0.20	0.0002	2.14	1.44–3.19
GI symptoms (absent versus present)	0.80	0.29	0.005	2.23	1.28–3.91
Other pain (absent versus present)	−0.60	0.22	0.007	0.55	0.36–0.85
Bone disease/non-vertebral fracture (absent versus present)	−2.22	1.02	0.03	0.11	0.02–0.80

**Table 5 cancers-15-03337-t005:** Number of patients reporting each symptom divided by those with previously identified precursor disease versus de novo myeloma. Grouped as the first presenting symptoms or abnormalities triggering an investigation and those present at diagnosis.

	Initial Symptoms or Abnormality Reported ^ɸ^					Total Symptoms or Abnormalities Reported at MM Diagnosis				
Patient Reported Symptom Categories	Precursor Disease (*n* = 149)		De Novo Myeloma (*n* = 813)			Precursor Disease (*n* = 149)		De Novo Myeloma (*n* = 813)		
	N	%	N	%	*p* *	N	%	N	%	*p* *
Back pain	13	9	288	35	0.003	30	20	334	41	0.003
Other pain (excl. back pain)	21	14	213	26	0.06	38	26	257	32	>0.99
Systemic symptoms	14	9	194	24	0.003	25	17	240	30	0.06
Anaemia (self-reported)	12	8	57	7	>0.99	30	20	83	10	0.03
Respiratory symptoms	3	2	56	7	>0.99	5	3	69	8	0.96
Gastrointestinal symptoms	3	2	50	6	>0.99	5	3	68	8	>0.99
Infection	4	3	38	5	>0.99	7	5	47	6	>0.99
Neurological symptoms	5	3	28	3	>0.99	9	6	44	5	>0.99
Abnormal blood results (suggestive of myeloma)	4	3	25	3	>0.99	7	5	44	5	>0.99
Bone disease and fracture (patient-reported)	1	1	23	3	>0.99	5	3	37	5	>0.99
Other abnormal blood results	9	6	12	1	0.06	13	9	19	2	0.01
Renal problems (self-reported)	1	1	20	2	>0.99	5	3	41	5	>0.99
Other symptoms—not specified	2	1	17	2	>0.99	3	2	20	2	>0.99
Bleeding/thrombosis	2	1	14	2	>0.99	3	2	21	3	>0.99
Lumps	2	1	7	1	>0.99	3	2	9	1	>0.99
Urological symptoms	0	0	6	1	>0.99	0	0	8	1	>0.99

* *p*-values with Bonferroni correction. ^ɸ^ The questionnaire was not designed to gain information about the initial symptoms or abnormalities triggering a referral for a precursor disorder.

**Table 6 cancers-15-03337-t006:** Comparison of the baseline and prognostic parameters at trial entry for those with previously identified precursor disease versus de novo multiple myeloma.

		Precursor Disease N = 149		De Novo Myeloma N = 813		
Factor	Grouping	N	%	N	%	*p* *
Age (years)	Median	70		67		0.06
	IQR	62–78		59–74		
Ethnicity	White	138	93	738	91	>0.99
Gender	Male	81	54	520	64	0.42
Planned high intensity treatment	Yes	69	46	454	56	0.56
PS 6 m before	0–1	136	91	742	91	>0.99
	2–4	6	4	44	5	
	Unavailable	7	5	27	3	
PS at entry	0–1	126	85	598	74	0.001
	2–4	17	11	203	25	
	Unavailable	6	4	12	1	
Decline in PS	No change	106	71	430	53	0.001
	Deteriorated	35	24	356	44	
	Unavailable	8	5	27	3	
ISS	1	39	26	174	21	0.04
	2	63	42	287	35	
	3	20	13	227	28	
	Unavailable	27	18	125	15	
Hypercalcaemia	Yes	2	1	43	5	>0.99
	No	114	77	594	73	
	Unavailable	33	22	176	22	
Renal impairment	Yes	18	12	130	16	>0.99
	No	128	86	672	83	
	Unavailable	3	2	11	1	
Anaemia	Yes	44	30	355	44	0.03
	No	76	51	303	37	
	Unavailable	29	19	155	19	
Bone disease	Yes	82	55	598	74	0.001
	No	67	45	208	26	
	Unavailable	0	0	7	<1	
Vertebral fracture	Present	27	18	243	30	0.001
	Absent	53	36	345	42	
	Unavailable	69	46	225	28	
Number of CRAB features	0	31	21	44	5	0.001
	1	61	41	309	38	
	2	23	15	216	27	
	3+	0	0	62	8	
	Unavailable	34	23	182	22	

* *p*-values with Bonferroni correction.

**Table 7 cancers-15-03337-t007:** Symptom profile for TEAMM trial participants compared with those reported in the literature and in suspected cancer guidelines.

	Symptom	N	Study Method/Symptom Recording	Back Pain	Other Pain: Generalised or Any Part of Body	Bone Pain	Chest Pain +/− Rib Pain	General Systemic Symptoms	Anaemia as Trigger for Referral	Tiredness or Fatigue	Night Sweats	Weight Loss	Infection	Unexplained Fracture or Bone Lesions	Short of Breath	Bleeding
Study Country																
TEAMM trial/UK		962	Questionnaire/ Patient own words -free text	38	31	4	10	28	12	16	1	9	6	4	6	3
Shephard 2015 */UK		2703	Primary care datalink (CPRD)/Predetermined GP symptom list	28	26 *Chest 15, joint 4, rib 3, combined bone 4*	4	18					4		6	10	3
Howell 2015 [21], *^,^°/UK		134	Questionnaire/Predetermined specialist symptom list			73.7		52.5		43.2	10.2	16.9	5.1		18.6	12.7
Howell 2013 [2] *^,µ^/UK		493	HMRN registry/ Patient own words -free text		46.2		2.4			21.1			6.4	11.3 *Joint problem/fracture*	6.8	
Forbes 2014 [8] *^,^°/UK		134	Questionnaire/ Predetermined specialist symptom list		65.7 *musculoskeletal, chest, abdominal*			47 *fatigue, weight loss, anorexia*							20.1 *SOB and cough*	6
Goldschmidt 2016 [4]/Israel		110	Insurance registry/predetermined symptom fields	58	34					25		9				
Kariyawasan 2007 [7]/UK		92	Retrospective healthcare record review			67		14 *asthenia*		9		5			5	
Kyle 2003/USA [20]		1027	Retrospective healthcare record review			58				32		24				
NICE guidelines for suspected cancer [23] ^ɸ^				yes	No	Yes		No	No	No	No	No	No	Yes		

Figures are all % of patients experiencing the symptom. GP = general practitioner/primary care physician. Table adapted from an original in the MD thesis by Catherine Atkin [24]. * These datasets include patients with smouldering myeloma, as well as active MM. ° Howell 2015 and Forbes 2014 were taken from the same dataset, but the symptoms subdivided differently. ^µ^ Howell 2013 provided additional data via personal communication. ^ɸ^ International searches for guidelines for suspected cancer were linked consistently to these guidelines [23].

## Data Availability

For the original data, please contact g.iqbal@warwick.ac.uk.

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
