# Peer review of "Presenting Symptoms in Newly Diagnosed Myeloma, Relation to Organ Damage, and Implications for Symptom-Directed Screening: A Secondary Analysis from the Tackling Early Morbidity and Mortality in Myeloma (TEAMM) Trial"

_cancers, 2023, doi:10.3390/cancers15133337_

Round 1

Reviewer 1 Report

The Authors reviewed presenting symptoms and their changes over time in 962 newly diagnosed multiple myeloma (MM) patients, among whom 149 had a precursor disease whereas 813 patients were de novo MM. The study emphasized that, despite symptom burden is high in MM patients, myeloma symptoms are not specific and the perception of them can be different between patient and healthcare worker. The study is original, interesting and well written although it suffers from a limit that is the median age of patients (67 years), younger than that expected in a real life population. Had patients other comorbidities that could justify symptoms reported in Table 2? Did the Authors try to construct a risk model using parameters that were significant at multivariate analysis? Despite limits, this paper may represent the basis for further studies that include also older patients

Author Response

Thank you for the review. In reply to the specific questions:

Reviewer 1: Had patients’ other comorbidities that could justify symptoms reported in Table 2?

Patients were specifically asked at baseline ‘When did the patient first notice bodily changes and/or symptoms that they attribute to the myeloma?’ (Table S2). This question, which aligns with the Aarhus statement, was an attempt to ask patients to discriminate symptoms that in retrospect were due to myeloma and not a comorbidity. Comorbidities were collected on a rating scale 1-mild, 2-moderate, 3-severe, 4-extremely severe.  There were only 19 patients in whom the comorbidity was scored 3 or 4 and was unrelated to myeloma: These were scattered 1-4 cases per organ system and being such small numbers did not influence the results. A sentence about comorbidities has been added to the manuscript.

Reviewer 1: Did the Authors try to construct a risk model using parameters that were significant at multivariate analysis?

A risk model based on symptoms and laboratory results has previously been published for primary care (9). Our analysis was on the relation between symptoms and irreversible organ damage.  Multivariate analysis revealed only age and 1 or 2 other parameters as significant in relation to VCFs or renal failure respectively. The authors did not think it necessary to construct a risk model as the manuscript already contained much analysis and might risk over-analysis.

Reviewer 2 Report

This article, which is really difficult to judge, focuses on the analysis of the symptoms of patients with MM in the pre-diagnosis phase. The complexity of the statistical analyzes makes it very difficult to understand the material and methods to the results, a circumstance that is further complicated by the inclusion of the tables in the body of the text, including table 2, which is misaligned and, at least for me, it is incomprehensible. I believe that the wording should be simplified to make it more accessible, and it would probably be easier to read if any of the tables were moved to supplementary material.

On the other hand, in addition to the description of the clinical development between the onset of symptoms and diagnosis in MM, a matter largely already acquaintance, on the other hand, reading throughout this article causes the reader to feel that the story is incomplete. That is, the significance of the delay in diagnosis with respect to early diagnosis, or in other words, the effect of the delay in starting treatment in more advanced stages of the disease. As far as I have been able to review, this aspect has not been explored in other publications of the TEAMM project. I do not understand why an analysis of survival and disease-free survival based on the delay in diagnosis and the category of the main symptoms observed is not included. It would also be useful to analyze the correlation of the symptoms collected in the analysis with the classification of MM by stage at diagnosis promulgated by the IMWG in 2015 (Palumbo A, et al. Revised International Staging System for multiple myeloma: a report from IMWG. J Clin Oncol. 2015;33:2863-2869).

In summary, the article offers data that may be of importance for the clinical management of MM, but the writing must be more accessible and the results, beyond their mere description, must be contextualized to improve the clinical interpretation of the results.  

Author Response

Thank you for your review. In reply to the first paragraph:

Table 2 has been moved to the supplement and the wording improved.

In reply to the second paragraph:

The trial only followed up patients for 12 months. Overall survival at 12 months was high at 91% and there was no benefit to small subgroup analysis. Prognostic factors at diagnosis were therefore used including the ISS staging system. Cytogenetic data were not collected in this trial and therefore the standard ISS rather than the revised ISS was used. A statement about the reasons for lack of overall survival data has been added to the manuscript.

Round 2

Reviewer 2 Report

I thank the authors for the changes they have made according to my suggestions. I must admit that none of the revisions in this article have been easy, basically due to the complexity of the statistical analysis. I believe, after several readings, however, and in conclusion, that the essence of the objectives and the results of the study should not be hidden by the difficulty and complexity of the statistics or the number of patients included in the analysis.

In my opinion, the problem with this study is that the clinical data that constitute the objective of this study have already been known for a long time. At this point in the History of Multiple Myeloma, what is the value of discovering that bone lesions and pathological fractures cause pain or back pain if the fractures affect the vertebrae?, or that these lesions cause pain and functional limitations and that patients describe "back pain" as "pain" and not as "bone pain"? Likewise, what is the point of investigating the widely known truism that a patient monitored by a hematologist due to a Monoclonal Gammopathy or Smoldering Multiple Myeloma has a greater chance of benefit from an early diagnosis, and even early treatment and therefore a better prognosis?.

These and other data that are analyzed do not contribute innovation to the knowledge of the disease. Proof of this is to verify that the results of this investigation appear in their entirety for a long time even in the general  Internal Medicine Treatises (see eg: Harrison's Manual of Medicine, 2020)